# KDM5 demethylases suppress R-loop-mediated 'viral mimicry' and DNA damage in breast cancer cells

Lena Lau[1], Kurt Henderson[1], Ahu Turkoz[1], Sara Linker[2], Dorte Schlessinger[3], Brad Townsley[4], Brian Egan[4], Shoba Ragunathan[5], Robert Rollins[2], Xianju Bi[6], Zhijian J Chen[6], Oleg Brodsky[2], Clifford Restaino[2], Murali Gururajan[2], Kristen Jensen-Pergakes[2], Anders Mälarstig[7], Chames Kermi[1†], Paul Moore[1†], Marie Classon[1*†]

[1]Pfizer Center for Therapeutic Innovation, San Francisco, United States; [2]Pfizer Oncology, La Jolla, United States; [3]Karolinska Institutet, Stockholm, Sweden; [4]Active Motif, Carlsbad, United States; [5]Pfizer, New York, United States; [6]The University of Texas Southwestern Medical Center, Dallas, United States; [7]Pfizer, Stockholm, Sweden

*For correspondence:
classon.marie@gmail.com

†These authors contributed equally to this work

## eLife Assessment

This study presents a **valuable** finding that KDM5 inhibitors may enable a wide therapeutic window as compared to STING agonists or type I interferons. The evidence supporting the claims of the authors is **convincing**. The work will be of broad interest to scientists working in the field of breast cancer research.

**Abstract** Tumors with low expression of interferon-stimulated genes (ISG) and antigen presentation (AP) genes respond relatively poorly to current immunotherapies. One of the early hallmarks of cancer is DNA hypomethylation in genomic repeat regions that can result in the expression of normally silenced endogenous 'viral' elements. Such epigenetic changes have the potential to augment anti-tumor immune responses as well as reduce tumor cell fitness through the generation of aberrant nucleic acid species (NAS) and consequent activation of NAS-sensing pathways. Therefore, tumor evolution should favor additional selective events that suppress NAS generation, possibly yielding specific therapeutic vulnerabilities. Here, we show that the lysine demethylase 5 (KDM5) family of epigenetic regulatory enzymes suppresses R-loop formation in genomic repeat regions specifically in cancer cells. We find that KDM5 inhibition in luminal breast cancer cells results in R-loop-mediated DNA damage, reduced cell fitness, and an increase in ISG and AP signatures as well as cell surface major histocompatibility complex (MHC) class I, mediated by RNA:DNA hybrid activation of the CGAS/STING pathway. KDM5 inhibition does not result in DNA damage or activation of the CGAS/STING pathway in normal breast epithelial cells, suggesting that KDM5 inhibitors may enable a wide therapeutic window in this setting, compared to STING agonists or type I interferons. These findings provide new insights into the interplay between epigenetic regulation of genomic repeats, R-loop formation, innate immunity, and cell fitness in the context of cancer evolution and therapeutic vulnerability.

## Introduction

Cancer treatments have evolved along three main paradigms: cytotoxic chemotherapy, rationally targeted agents, and immune-based therapies. Still, innate and acquired drug resistance remains key limitations to improving outcomes in advanced cancer patients. With immune-based therapies,

it is well documented that responses depend on an appropriate tumor microenvironment (TME). In recent years, it has been shown that tumor-intrinsic expression of endogenous, virus-like elements can affect both tumor cell viability and the TME by mimicking viral infection and activating pattern recognition receptors (PRRs) such as TMEM173/STING and melanoma differentiation-associated protein 5 (MDA5) – a process coined 'viral mimicry' (*Roulois et al., 2015*; *Chiappinelli et al., 2015*; *Chen et al., 2021*; *Jansz and Faulkner, 2021*). It is conceivable that suppression of the expression of these repeat elements and 'viral mimicry' may also mediate drug resistance to other therapies (*Guler et al., 2017*).

Throughout evolution, the human genome has been modified by waves of viral insertions, and a large fraction of the human genome is consequently composed of repetitive elements and retrotransposons, including interspersed long-terminal repeat (LTR)-based endogenous retroviruses (HERVs), non-LTR-based short- and long-interspersed nuclear elements (SINEs and LINEs), as well as centromeric, telomeric, and satellite repeats (*Lander et al., 2001*; *Wells and Feschotte, 2020*). HERV, LINE, and SINE elements are mostly silenced in healthy human somatic cells but can be deregulated in cancer cells due to DNA hypomethylation in regions containing such elements (*Feinberg and Vogelstein, 1983*; *Ehrlich, 2009*). Notably, aberrant expression of these elements is increasingly recognized as playing important roles, not only in cancer evolution and drug resistance but also in the development of other diseases (*Burns, 2020*).

Mechanistically, it has been shown that expression of HERVs, LINEs, and SINEs can cause cells to appear virally infected to the immune system (termed 'viral mimicry'), in part through the formation of double-stranded RNAs (dsRNAs) that are detected by MDA5, thereby triggering a type I interferon (IFN-I) response that can decrease tumor cell fitness and increase immunogenicity. Consequently, during tumor evolution such a response may need to be blunted by the acquisition of mutations in genes that promote IFN-I signaling, or by the possible deployment of viral restriction factors or compensatory epigenetic repressive mechanisms (*Chen et al., 2021*; *Vashi and Bakhoum, 2021*; *Cheon et al., 2023*). Interestingly, the latter two non-mutational mechanisms may serendipitously present unique tumor-specific therapeutic opportunities. For example, it has been shown that the A-to-I-editing viral restriction factor ADAR (*Nishikura, 2016*) destabilizes inverted repeat SINE elements (IR-Alus) and prevents their detection by MDA5 (*Mehdipour et al., 2020*). As a result, tumors that harbor increased expression of SINE elements (and consequently ISGs) are highly sensitive to ADAR loss (*Gannon et al., 2018*). Notably, ADAR disruption has also been shown to lead to both intrinsic anti-tumor activity and enhanced immunotherapy response in preclinical mouse models (*Ishizuka et al., 2019*; *Dubrot et al., 2022*). Previous studies have also demonstrated that regulators of DNA methylation (*Roulois et al., 2015*; *Chiappinelli et al., 2015*; *Wu et al., 2024*), as well as inhibitors of other epigenetic regulators such as histone methyltransferases (HMTs) (*Shen et al., 2021a*; *Cuellar et al., 2017*; *Morel et al., 2021*; *Deblois et al., 2020*) and histone lysine demethylases (KDMs) (*Sheng et al., 2018*; *Wu et al., 2018*; *Zhang et al., 2021*; *Shen et al., 2021b*; *Leadem et al., 2018*), can induce 'viral mimicry' by reactivating repeat elements and increasing ISG signatures beyond a threshold level of tolerance, thereby affecting cell survival.

Notably, suppression of repeat transcription may also be responsible for the low levels of MHC class I on the surface of tumor cells (*Dhatchinamoorthy et al., 2021*). Reduced surface MHC I on tumor cells will decrease their intrinsic immunity but also limit responsiveness to therapies such as adoptive T cell therapies. In addition, peptides derived from the repeat genome have also been shown to be presented by tumor cells (*Saffern and Samstein, 2023*; *Griffin et al., 2021*; *Bonté et al., 2022*; *Kong et al., 2019*).

In addition to dsRNA activation of MDA5, viral mimicry can also be mediated through the CGAS/STING pathway, which is initiated by the sensing of DNA or RNA:DNA hybrid species (*Chen et al., 2016*; *Mankan et al., 2014*). Agonists of the STING pathway are currently being evaluated in clinical trials. The lysine demethylase 5 family (KDM5A-D), which removes histone 3 lysine 4 (H3K4) di- and tri-methyl modifications (*Pavlenko et al., 2022*), has been suggested to blunt 'viral mimicry' through the repression of STING expression or modification of signaling components downstream of STING (*Wu et al., 2018*; *Zhang et al., 2021*; *Shen et al., 2021b*). Here, we demonstrate that disruption of KDM5 activates 'viral mimicry' in luminal breast cancer cell lines that are responsive to STING agonists. We also show that KDM5 inhibition specifically promotes 'viral mimicry' in tumor cells and not in normal primary breast epithelial cells.

Mechanistically, we demonstrate that inhibition of KDM5 and the resulting increase in H3K4 tri-methylation in repeat regions leads to an increase in R-loop formation. R-loops are three-stranded nucleic acid structures that naturally form during cellular processes such as transcription, replication, and DNA repair when a nascent RNA transcript hybridizes to its DNA template, leaving a loop of single-stranded DNA (*Crossley et al., 2019*). Although R-loops are crucial intermediates in normal cellular processes, if not properly regulated, they can also compromise genomic stability, induce DNA damage, and trigger inflammatory responses through upregulation of IFN signaling. As such, dysregulation of R-loops and the cGAS/STING pathway has been implicated in both cancer and autoimmune diseases (*Mackay et al., 2020*; *Cristini et al., 2022*). As an example, it has recently been demonstrated that disruption of the RNA:DNA helicase senataxin (SETX), which can resolve R-loops, in cancer cells results in accumulation of R-loop-derived cytoplasmic RNA:DNA hybrids that can activate an innate immune response (*Crossley et al., 2023*). Increases in transcription-replication conflicts (TRCs) and subsequent R-loop formation have also been linked to DNA damage and heightened sensitivity to PARP inhibitors (*Petropoulos et al., 2024*; *Laspata et al., 2023*; *Kemiha et al., 2021*). Here, we show that KDM5 disruption not only stimulates STING-mediated ISG and AP signatures as well as MHC class I presentation but also induces a DNA damage response that is independent of the cGAS/STING-type I IFN axis. In summary, our studies suggest that KDM5 inhibitors hold promise as stand-alone cancer therapies or in combination with immune or DNA-damaging cancer therapies.

## Results

### Luminal breast cancer cell lines display features of 'cold' tumors, but "viral mimicry" can be induced through the STING, MAVS, and IFN-I pathways

Immune-directed therapies can produce significant clinical benefit for some, but not all, cancer patients. Breast cancers are among the tumor types that are least responsive to immune-based therapies and display low ISG and AP gene expression signatures compared to tumor types that respond well to such agents. Among the breast cancer molecular subtypes, luminal breast cancers not only display the lowest ISG and AP gene signatures (*Figure 1a–c*, *Figure 1—figure supplement 1a and b*, and *Supplementary file 1*) but also exhibit a lower number of MHC class I molecules on their surface in comparison to triple-negative breast cancer cell lines (*Figure 1d*), as well as tumor types that are more immunogenic (data not shown). Collectively, these features are indicative of 'cold' tumors and most likely contribute to the poor response to immune-based therapies seen for this breast cancer subtype.

As mentioned above, loss of DNA methylation in genomic repeat regions of cancer cells is an early event in tumor development that potentially contributes to mutations, translocations, and expression of oncogenes. However, de-repression of LINE, SINEs, HERVs, and other repeat elements also promotes 'viral mimicry': generation of cytoplasmic NAS and activation of PRRs and IFN-I signaling pathways. Therefore, during tumor evolution, various mechanisms can be engaged to counter the negative effects of 'viral mimicry' and create an equilibrium that favors the tumor (*Kermi et al., 2022*). To better understand the 'viral mimicry' state in luminal breast cancer cells, we first examined a panel of cell lines for their ability to respond to PRR agonists or to IFN-I stimulation (see *Figure 1e* for a simplified schematic of normal PRR signaling). Using MHC class I surface levels as well as activation of an integrated interferon-stimulated response element (ISRE) reporter as read-outs, these studies demonstrate that all tested luminal breast tumor cell lines respond to IFNβ (a type I IFN) to varying degrees (*Figure 1f*, upper panels and *Figure 1g*, left panel, as well as *Figure 1—figure supplement 1c*, left panel, and *Figure 1—figure supplement 1e*, upper panel). Most of the luminal lines also respond to MDA5/RIGI agonists, such as Poly-IC, or an in vitro transcribed inverted repeat SINE element (IR-Alu) (*Figure 1f*, lower panel, and *Figure 1g*, right panel, as well as *Figure 1—figure supplement 1d*, right panel, and *Figure 1—figure supplement 1e*, bottom two panels), but demonstrate a varied response to the STING agonist di-ABZI (*Figure 1f*, middle panel, and *Figure 1g*, middle panel, as well as *Figure 1—figure supplement 1c*, right panel, and *Figure 1—figure supplement 1e*, second panel). Collectively, these data show that most of the luminal breast tumor cell lines in this panel have intact signaling pathways downstream of PRRs, suggesting that PRR agonists could be used to increase the ISG and AP signatures and decrease cell viability in luminal breast cancer cells. However, like type I

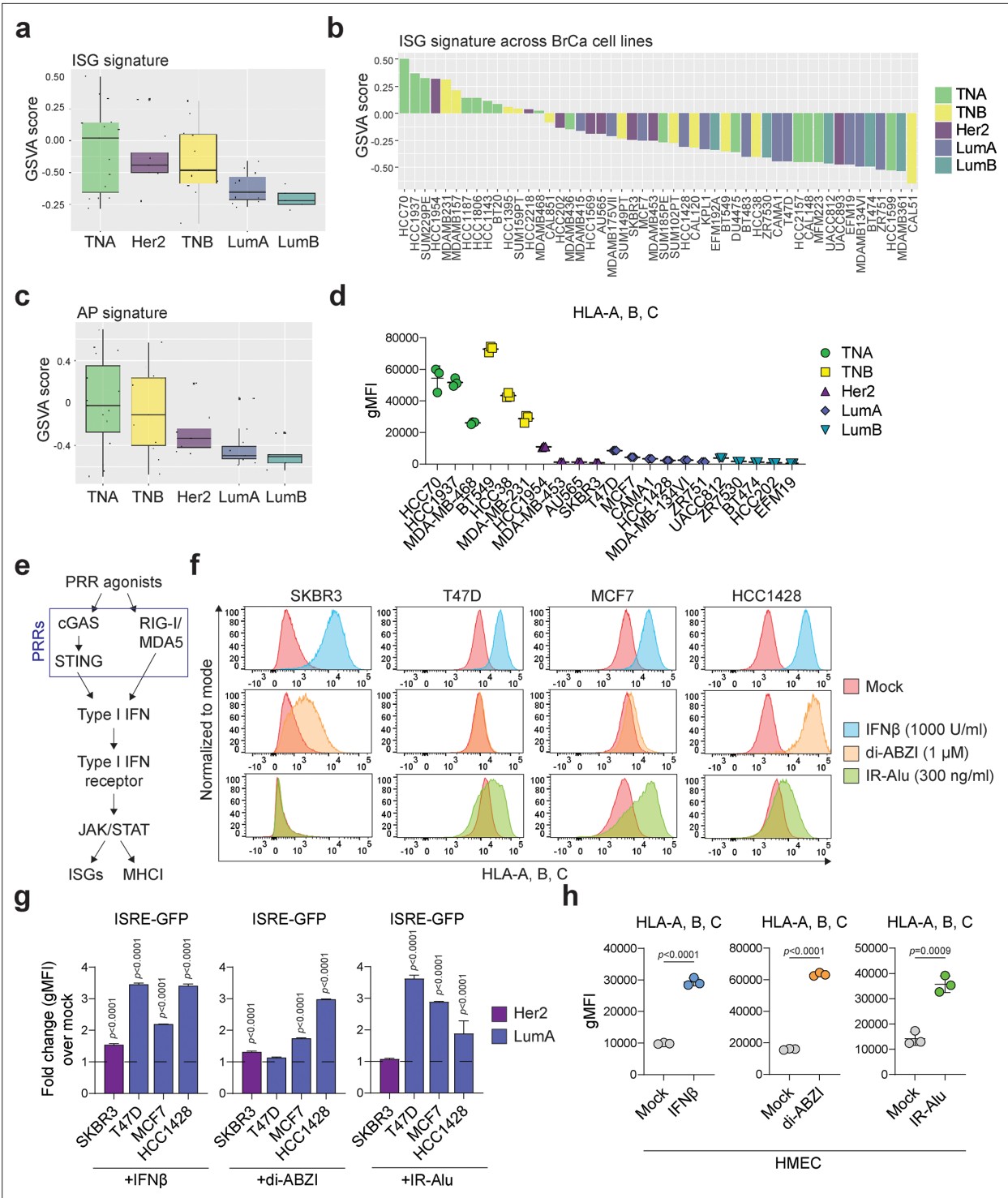

**Figure 1.** Low immune signatures in luminal breast cancer cells can be increased by agonists that activate type I IFN. (**a**) Gene set variation analysis (GSVA) enrichment scores of IFN-stimulated gene (ISG) set expression in breast cancer cell lines grouped by subtype and (**b**) across individual breast cancer cell lines. TNA/B: triple-negative A/B; Her2: Her2-positive; LumA/B: luminal A/B. (**c**) GSVA enrichment scores of antigen presentation (AP) gene set expression in breast cancer cell lines grouped by subtype. (**d**) Quantification of HLA-A, B, C surface levels in the indicated breast cancer cell lines, measured as geometric mean fluorescence intensity (gMFI) by flow cytometry. n=3 for each cell line. (**e**) Simplified schematic of type I IFN signaling activation by activation of pattern recognition receptors (PRRs). Parts of this pathway may be disrupted in cancer cells. (**f**) Flow cytometry histograms of HLA-A, B, C surface levels in indicated cell lines following treatment as indicated for 24 hours. Plots are representative of three separate experiments. (**g**) FACS quantification of ISRE-GFP fluorescence in indicated cell lines, colored by subtype, following treatment as indicated for 24 hours. Data are

*Figure 1 continued on next page*

*Figure 1 continued*

represented as fold change over mock-treated control. n=3, data are mean ± sd, p-values are from two-way ANOVA using Sidak's multiple comparisons, comparing each treatment vs mock treatment in all cell lines. (**h**) gMFI of HLA-A, B, C surface levels in HMEC cells, as determined by flow cytometry. Cells were treated with 1000 U/ml IFNβ, 1 µM di-ABZI, or transfected with 300 ng/ml IR-Alu for 24 hours. n=3, data are mean ± sd, p-values are from unpaired *t*-tests.

The online version of this article includes the following figure supplement(s) for figure 1:

**Figure supplement 1.** Low immune signatures in luminal breast cancer cells can be increased by agonists that activate type I IFN.

IFNs, PRR agonists are likely to indiscriminately induce such phenotypes in both tumor and normal epithelial cells, thereby limiting the therapeutic window for such agents. This is exemplified by an increase in MHC class I surface expression on both tumor cells and normal human mammary epithelial cells (HMECs) in response to STING or RIGI/MDA5 agonists as well as to type I IFN (*Figure 1h*).

Next, we evaluated factors shown to be involved in upstream suppression of STING or MAVS signaling in human tumor cell lines or in suppression of tumor immune responses in mouse models (*Dubrot et al., 2022*, *Figure 2—figure supplement 1a* for summary). These included the A-to-I editing enzyme ADAR1 and the histone demethylase KDM5, implicated in suppressing signaling through MDA5/MAVS and STING, respectively (*Mehdipour et al., 2020*; *Wu et al., 2018*). The HCC1428 luminal breast cancer cell line was chosen as a model for most of these studies because it responds well to STING and MDA5 agonists as well as to IFN-I (*Figure 1f and g*, *Figure 1—figure supplement 1c–e*). Consistent with the effects of MDA5 agonists, CRISPR-mediated disruption of ADAR1 (both p110 and p150 subunits) in HCC1428 cells (*Figure 2—figure supplement 1b*) results in an increase in both ISRE-reporter activity (*Figure 2a*, upper panel) and surface presentation of MHC class I molecules (*Figure 2a*, lower panel). Similar results were also observed in T47D and MCF7 luminal breast cancer lines (data not shown). Likewise, the KDM5-specific inhibitor C48 (*Liang et al., 2017*) increases ISRE reporter expression (*Figure 2b*, upper panel) as well as ISG and AP signatures (*Figure 2c*) in HCC1428 cells. MHC class I surface expression is also increased following C48 exposure in HCC1428 cells (*Figure 2b*, lower panel) and other luminal breast cell lines (*Figure 2—figure supplement 1f* and data not shown). As previously shown in other cell lines (*Wu et al., 2018*), inhibition of KDM5 results in 'viral mimicry' activation primarily mediated through the cGAS/STING pathway (*Figure 2d and e*, *Figure 2—figure supplement 1c–f*), and most likely not through the MAVS pathway (*Figure 2—figure supplement 1g and h*). Consistent with other studies (*Wu et al., 2018*), KDM5-dependent effects on viral mimicry in HCC1428 cells are primarily mediated by KDM5C (*Figure 2—figure supplement 1i and k*). We also observe that the effect on surface levels of MHC class I is mostly mediated by the KDM5C paralog in the HCC1428 cell line model (*Figure 2—figure supplement 1j*).

To evaluate whether viral mimicry activation and the subsequent increase in surface MHC class I can affect T-cell receptor (TCR)-driven T cell activation, we developed an MCF7:Jurkat T cell co-culture system (*Figure 2f* and 'Materials and methods'). Briefly, luminal MCF7 breast cancer cells engineered to over-express the NY-ESO1 antigen were cultured with Jurkat cells expressing a NY-ESO1-specific TCR and a luciferase reporter under control of the nuclear factor of activated T cells (NFAT) promoter that is activated upon TCR activation. Using this system, we demonstrated that disruption of ADAR in MCF7:NY-ESO1 cells results in robust activation of the NFAT reporter in co-cultured Jurkat:NY-ESO1 TCR cells (*Figure 2g*). Treatment with the KDM5 inhibitor C48 prior to co-culturing these cells with NY-ESO1 TCR Jurkat cells also results in a dose-dependent increase in NFAT reporter activity (*Figure 2h*)>The activation of the NFAT reporter by KDM5 inhibition was reversed upon cGAS or STING depletion in MCF7:NYESO1 cells (*Figure 2i*). Analogous to the effects seen on ISG and AP signatures, induction of the Jurkat NFAT reporter is largely dependent on the disruption of the KDM5B and C paralogs in the MCF7:NY-ESO1 cell line (*Figure 2j*). Taken together, these experiments demonstrate that PRR and IFN-I signaling can be augmented in luminal breast cancer cells by direct agonists or by the disruption or inhibition of factors that may repress these pathways upstream.

## The effect of KDM5 inhibition on cell fitness is not mediated through activation of cGAS/STING or type I IFN signaling

In addition to increasing the immunogenicity of tumor cells, 'viral mimicry' can also reduce cell fitness. This relationship is well-documented in the MDA5/MAVS dsRNA-sensing pathway. In brief, MDA5/

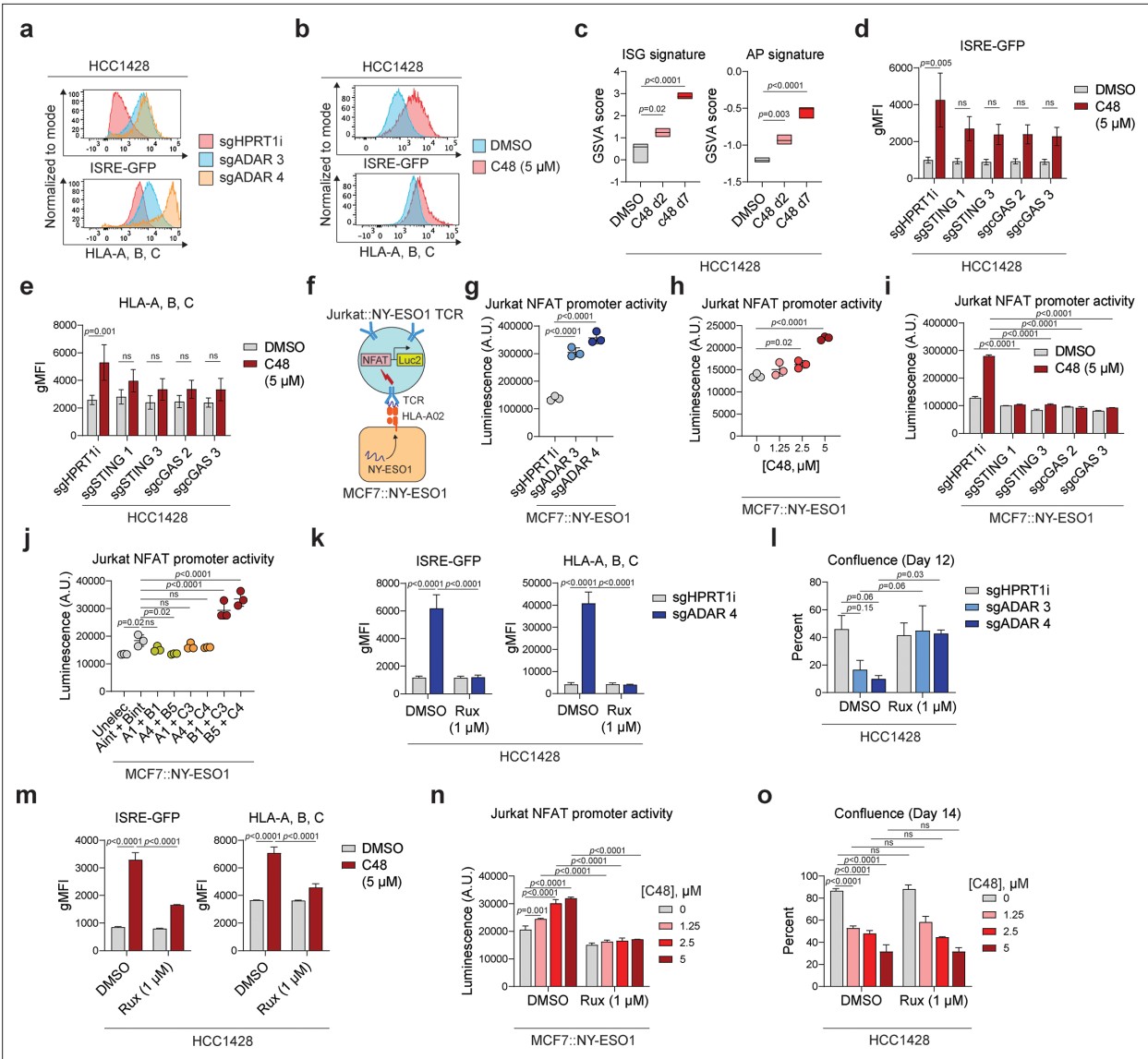

**Figure 2.** Disruption of upstream regulators of PRR signaling activates viral mimicry through distinct mechanisms. (**a**) Representative flow cytometry histograms of ISRE-GFP fluorescence and HLA-A, B, C surface expression (geometric mean fluorescence intensity, gMFI, normalized to mode) in CRISPR disrupted HCC1428 control (sgHPRT1i) or ADAR (sgADAR) cells. (**b**) Representative flow cytometry histograms of ISRE-GFP fluorescence and HLA-A, B, C surface expression (gMFI normalized to mode) in HCC1428 cells treated mock or C48 treated as indicated for 7 days. (**c**) Gene set variation analysis (GSVA) enrichment scores of ISG and AP gene set expression in HCC1428 cells treated with DMSO for 7 days as a control, or 2.5 μM C48 for 2 or 7 days as indicated. n=3, data are derived from RNAseq analysis, boxplots show min to max, line at median. p-Values are from ordinary one-way ANOVA using Dunnett's multiple comparisons test. (**d**) gMFI of ISRE-GFP activity and (**e**) HLA-A, B, C surface levels, as determined by flow cytometry, in HCC1428 cells CRISPR disrupted using an intronic cutting control (sgHPRT1i) or two independent sgRNAs against STING or cGAS and treated as indicated. Samples were harvested 10 days post-electroporation and 7 days post-treatment. n=3, data are mean ± sem, p-values are from two-way ANOVA using Sidak's multiple comparisons test. (**f**) Schematic of MCF7/Jurkat co-culture system. (**g**) Luminescence from NFAT-luciferase reporter (in arbitrary units A.U.) of engineered Jurkat cells co-cultured with engineered MCF7 cells: 4 days post-CRISPR disruption with the indicated sgRNAs, (**h**) 7 days post-treatment with mock and the indicated concentrations of C48, (**i**) 10 days post CRISPR disruption with the indicated sgRNAs and treated as indicated for 7 days, and (**j**) 7 days post-electroporation with sgRNAs against two of three KDM5 paralogs. Luciferase activity was measured 5 hours after co-culture, n=3, data are mean ± sd, p-values are ordinary one-way ANOVA using Dunnett's multiple comparisons test in (**g–h**) ordinary one-way ANOVA using Sidak's multiple comparisons test in (**i**), and two-way ANOVA using Tukey's multiple comparisons test in (**j**). (**k**) Quantification of ISRE-GFP activity and HLA-A, B, C surface expression, measured as gMFI, in HCC1428 cells 7 days post-CRISPR disruption using sgRNAs against intronic HPRT1 or ADAR as indicated. Cells are treated with ruxolitinib (Rux) as indicated throughout the 7 days of the experiment. n=3, data are mean ± sd, p-values are from two-way ANOVA using Tukey's multiple comparisons test. (**l**) Percent confluence of HCC1428 cells 12 days post-CRISPR disruption using sgRNAs against intronic HPRT1 or ADAR. Cells are treated with DMSO or Rux as indicated throughout the 12 days. n=3, data are mean ± sem, p-values are from two-way ANOVA using Sidak's multiple comparisons test. (**m**) Quantification of ISRE-GFP and HLA-A, B, C surface levels, determined by flow cytometry

*Figure 2 continued on next page*

*Figure 2 continued*

and measured as gMFI, in HCC1428 cells treated as indicated for 7 days. n=3, data are mean ± sd, p-values calculated as in (**k**). (**n**) Luminescence of engineered Jurkat cells co-cultured with MCF7:NY-ESO1 treated as indicated for 7 days, measured 5 hours after co-culture. n=3, data are mean ± sd, p-values are from two-way ANOVA using Tukey's multiple comparisons test. (**o**) Percent confluence of HCC1428 cells 14 days post-treatment as indicated. Samples were split at the same ratios on day 7 of treatment. n=3, data are mean ± sd, p-values are from two-way ANOVA using Tukey's multiple comparisons test.

The online version of this article includes the following source data and figure supplement(s) for figure 2:

**Figure supplement 1.** Perturbation of upstream regulators of PRR signaling activates viral mimicry through distinct mechanisms.

**Figure supplement 1—source data 1.** Raw, unlabeled blot images corresponding to panel (b).

**Figure supplement 1—source data 2.** Raw blot images corresponding to panel (b) with target proteins labeled.

**Figure supplement 1—source data 3.** Raw, unlabeled blot images corresponding to panel (c).

**Figure supplement 1—source data 4.** Raw blot images corresponding to panel (c) with target proteins labeled.

**Figure supplement 1—source data 5.** Raw, unlabeled blot images corresponding to panel (i).

**Figure supplement 1—source data 6.** Raw blot images corresponding to panel (i) with target proteins labeled.

MAVS activation by dsRNA induces IFN-I signaling through the JAK/STAT pathway, resulting in ISG induction. This includes an increase in Protein Kinase R (PKR), which binds dsRNA and signals through eIF2α to attenuate global translation and drive apoptosis (see *Figure 2—figure supplement 1l* for schematic). Consistent with this mechanism of action, treatment with ruxolitinib, a Janus Kinase (JAK) inhibitor that blocks the IFN-I response, rescues the effects of ADAR1 disruption not only on ISG induction, MHC class I presentation, but also overall tumor cell fitness (*Figure 2k–l*, *Figure 2—figure supplement 1m*). This direct relationship between ADAR1, dsRNA levels, IFN signaling, and cell fitness is further illustrated by the observation that cancer cell dependencies on ADAR, as reported in DepMap, correlate with baseline ISG and AP signatures (*Figure 2—figure supplement 1n* for summary; *Gannon et al., 2018*).

In contrast, ruxolitinib treatment does not reverse KDM5 inhibitor-mediated loss of cell fitness (*Figure 2o*) despite blunting induction of ISG/AP signatures and activation of the NFAT reporter in the MCF7:NYESO1/Jurkat co-culture system (*Figure 2m and n*, *Figure 2—figure supplement 1o and p*). Similarly, while genetic disruption of cGAS or STING upstream of IFN-I signaling abrogates ISG induction (*Figure 2d and e*, *Figure 2—figure supplement 1c–f*), it does not reverse C48-mediated cell fitness loss in HCC1428 cells (*Figure 2—figure supplement 1q*). In summary, these data demonstrate that KDM5, unlike ADAR1, regulates cancer cell fitness and ISG/AP signature induction ('viral mimicry') through distinct mechanisms.

## KDM5 inhibition induces viral mimicry and a type I IFN response in luminal breast cancer cells, but not in normal breast epithelial cells

Upstream regulators of PRRs may more selectively affect tumor cells compared to direct PRR agonists. For example, sensitivity to ADAR disruption is observed in tumors that exhibit relatively high ISG signatures (15 and *Figure 2—figure supplement 1n*), potentially yielding a therapeutic window. To investigate whether KDM5 inhibition would provide a larger therapeutic window than STING agonists, we compared the effect of C48 on ISG and AP signatures as well as cell fitness phenotypes in breast tumor cells and normal human mammary epithelial cells (HMECs). Although KDM5 inhibition increases H3K4me3 and cGAS/STING levels in both HCC1428 tumor cells and HMECs (*Figure 3a and b*, *Figure 3—figure supplement 1a*), treatment of HMECs with increasing concentrations of C48 did not induce a dose-dependent increase of MHC class I surface expression in HMECs as compared to luminal breast tumor lines (*Figure 3c*, *Figure 3—figure supplement 1b–c*). Furthermore, KDM5 inhibition did not elevate ISG and AP signatures in HMECs as compared to HCC1428 cells, or other KDM5 inhibitor-sensitive luminal breast cancer cells (*Figure 3d*, *Figure 3—figure supplement 1d–f* and data not shown). Together, these data show that the effect of KDM5 inhibition on ISG and AP signatures is tumor cell-specific in this tissue and not solely dependent on increased STING levels as previously suggested (*Wu et al., 2018*).

The STING-independent, adverse effects of KDM5 disruption on cell fitness are also specific to tumor cells (*Figure 3e*, *Figure 3—figure supplement 1g and m*) and are largely dependent on KDM5C (*Figure 3—figure supplement 1h and i* and data not shown). Interestingly, the expression

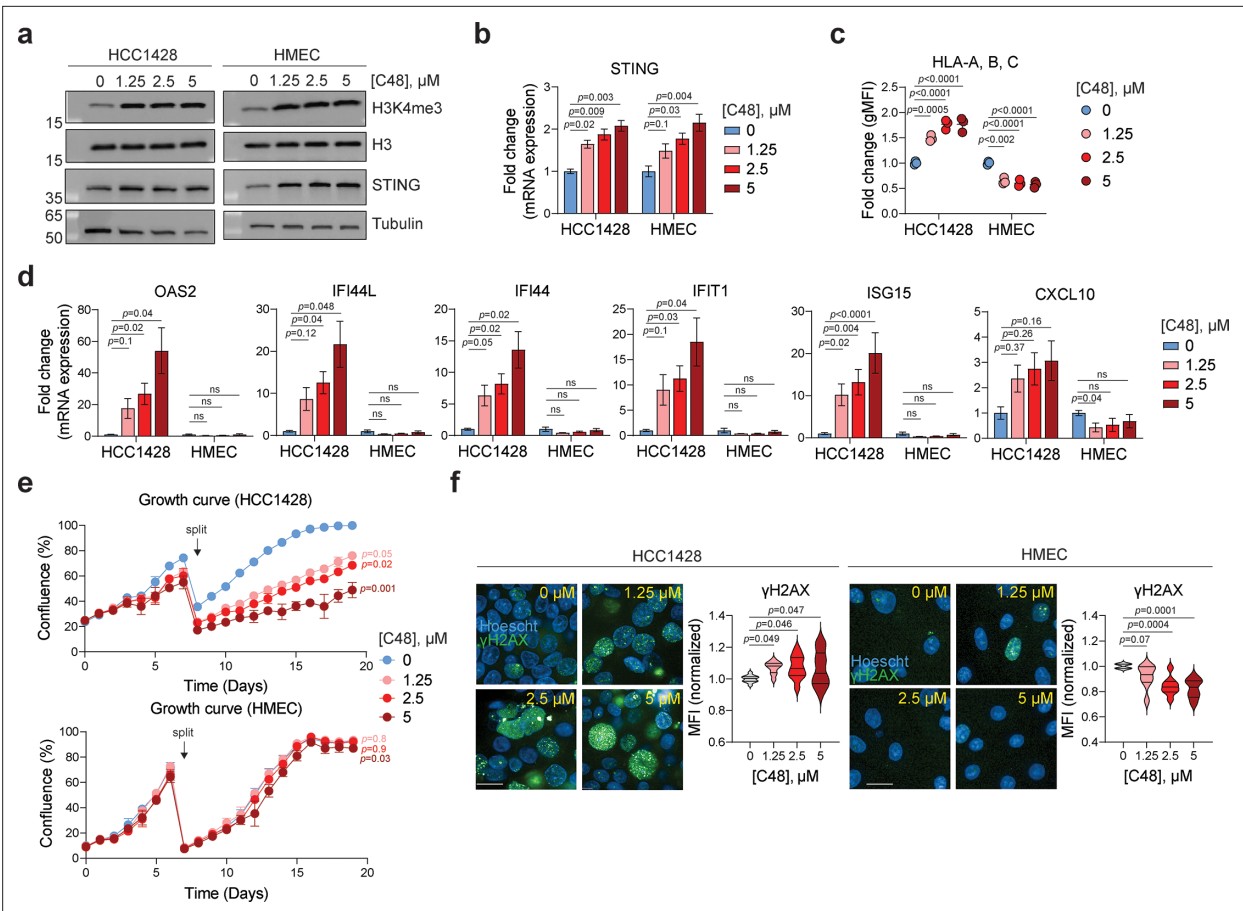

**Figure 3.** KDM5 inhibition induces 'viral mimicry' and DNA damage specifically in tumor cells, but not in normal epithelial breast cells. (**a**) Representative immunoblot of the indicated proteins in HCC1428 tumor cells and normal HMEC cells treated with mock or the indicated concentrations of C48 for 7 days. Histone H3 and tubulin were used as loading controls. (**b**) qRT-PCR analysis of STING mRNA expression levels, measured as 2^(Ct values of indicated gene subtracted from Ct value of β-actin), in normal and tumor cells treated as indicated for 7 days. Data are represented as fold change expression over DMSO-treated (0 μM) control. n=3–5, data are mean ± sem, p-values are from repeated measures one-way ANOVA using Dunnett's multiple comparisons test. (**c**) Quantification HLA-A, B, C surface levels, determined by flow cytometry and measured as geometry mean fluorescence intensity (gMFI), in normal and tumor cells treated as indicated for 7 days. Data are represented as fold change gMFI over DMSO-treated (0 μM) control. n=3, data are mean ± sd, p-values are from ordinary one-way ANOVA using Dunnett's multiple comparisons test. (**d**) qRT-PCR analysis of ISG signature genes from cells treated as indicated for 7 days. Data are analyzed as in (**b**). (**e**) Percent confluence of HCC1428 and HMEC cells treated with DMSO or the indicated concentrations of C48 over 19 days. Samples were split at the same ratios on day 7 of treatment. n=3, data are mean ± sd, p-values are from ordinary one-way ANOVA using Tukey's multiple comparisons test, comparing the calculated doubling time of the first 7 days derived from each curve to that of DMSO-treated (0 μM) control. (**f**) Representative immunofluorescence (IF) images and quantification of γ-H2AX staining (green) in cells treated with mock or the indicated concentrations of C48 for 7 days. Blue = Hoescht. Scale bar = 50 μM. Violin plots (with lines depicting quartiles) are derived from mean fluorescence intensity (MFI) of γ-H2AX staining, normalized to DMSO-treated (0 μM) control. n=8–13 wells, p-values are from mixed effects analysis using Dunnett's multiple comparisons test for HCC1428, repeated measures one-way ANOVA using Dunnett's multiple comparisons test for HMEC.

The online version of this article includes the following source data and figure supplement(s) for figure 3:

**Source data 1.** Raw, unlabeled blot images corresponding to panel (a).

**Source data 2.** Raw blot images corresponding to panel (a) with target proteins labeled.

**Figure supplement 1.** KDM5 inhibition or disruption induces viral mimicry and loss of cell fitness in luminal breast cancer cell lines.

**Figure supplement 2.** KDM5 inhibition induces ISGs in M2-polarized macrophages.

of KDM5B and C paralogs is somewhat higher in luminal breast cancer subtypes compared to triple-negative breast cancer subtypes (*Figure 2—figure supplement 1j–l*), with the latter also displaying higher ISG and AP signatures (*Figure 1a–d*). These data sets suggest that tumor cells with increased KDM5B and C expression display a lower ISG signature. This further strengthens the observations that disruption of KDM5B and C increases ISG and AP signatures in luminal breast cancer cell lines.

KDM5 paralogs have been suggested to play a role in DNA damage and repair (DDR) (*Gong et al., 2017*; *Gaillard et al., 2021*). To investigate whether a DDR phenotype contributes to the reduced cell fitness phenotype observed following KDM5 inhibition, we probed for the presence of gamma (γ)-H2AX as a measure of DNA damage. These experiments showed that γ-H2AX staining increases in the HCC1428 tumor cells, but not in HMECs, following exposure to C48 (*Figure 3f*). Taken together with the data presented in *Figure 2* suggesting that the effect of KDM5 inhibition on cell fitness is independent of STING and IFN-I, these results suggest that the cell fitness phenotype is driven by activation of a DNA damage response rather than an IFN-I response. Collectively, these data show that disruption of upstream regulators of PRR signaling, such as KDM5, can selectively affect tumor cells, thereby perhaps providing an increased therapeutic window for cancer treatment.

## KDM5 inhibition induces R-loop formation in areas of the repetitive genome that harbor increases in H3K4me3

As mentioned, previous studies have suggested that disruption of KDM5 family members activates the STING pathway partially by upregulating the expression of STING itself (*Wu et al., 2018*). However, our experiments show that STING levels are induced by KDM5 inhibition in both normal and tumor cells (*Figure 3b*), suggesting that increased levels of STING cannot be the only mechanism by which KDM5 disruption results in induction of innate immune phenotypes. Furthermore, our experiments also suggest that the effect of KDM5 inhibition or disruption on cell fitness is not mediated through STING or IFN-I signaling (*Figure 2o*, *Figure 2—figure supplement 1q*), but rather through a more complex mechanism involving activation of a DNA damage response.

Deregulated control of genomic R-loops (*Crossley et al., 2019*; *Mackay et al., 2020*) has been shown to result in DNA damage and stimulation of the cGAS/STING pathway via excised RNA:DNA hybrids (*Crossley et al., 2023*; *Figure 4a* for schematic). While R-loops are crucial regulators of normal biological processes, if not properly regulated, they can also pose a threat to genomic integrity; thus, R-loops represent a potential target for novel cancer therapeutics (*Elsakrmy and Cui, 2023*). In order to evaluate whether accumulation of R-loops can explain the phenotypes observed following KDM5 disruption in luminal breast cancer cells, we first stained control or C48-treated cells with GFP-tagged RNaseH1 protein, which selectively binds RNA:DNA hybrids (*Crossley et al., 2021*). These experiments show that KDM5 inhibition results in an increase of GFP+ cells, indicative of increased RNA:DNA hybrid formation in HCC1428 cells (*Figure 4b* for representative images and quantification). This increase is not observed in HMEC cells (data not shown). Furthermore, a partial depletion of XPF, an endonuclease involved in nucleotide excision repair that facilitates the excision of RNA:DNA hybrids from R-loops (*Crossley et al., 2023*), attenuates C48-mediated induction of 'viral mimicry', as measured by ISRE-GFP reporter activity and ISG mRNA expression in HCC1428 cells (*Figure 4c–e*, *Figure 4—figure supplement 1a*, left panel). Partial loss of XPF also attenuates the dose-dependent increase of MHCI pathway gene expression and surface levels induced by C48 (*Figure 4d*, right panel, *Figure 4f*, and *Figure 4—figure supplement 1a*, right panel). Collectively, these data suggest that the ISG and AP signatures induced by KDM5 inhibition in sensitive breast cancer cell lines are initiated by an increase in the generation of cytoplasmic R-loop-derived RNA:DNA hybrid species.

Transcription replication conflicts (TRCs), often magnified in cancer, can occur when the processes of DNA replication and transcription co-occur on the same DNA template and are intricately linked to the initiation/persistence of R-loops (*Goehring et al., 2023*). Transcriptional output is linked to epigenetic modifications, and it has previously been shown that the de novo DNA methyltransferase DNMT3b and other epigenetic regulators may mechanistically restrict R-loop-mediated DNA damage (*Shih et al., 2022*; *Bayona-Feliu et al., 2023*). Increased H3K4 methylation has been linked to increases in transcriptional output (*Wang et al., 2023*), possibly creating an epigenetic environment conducive to an increase in R-loops in tumor cells that already display an increase in TRC.

To explore a possible relationship between C48-induced increases in H3K4 trimethylation and genomic R-loops, we first performed Cleavage Under Targets and Tagmentation (CUT-and-Tag)

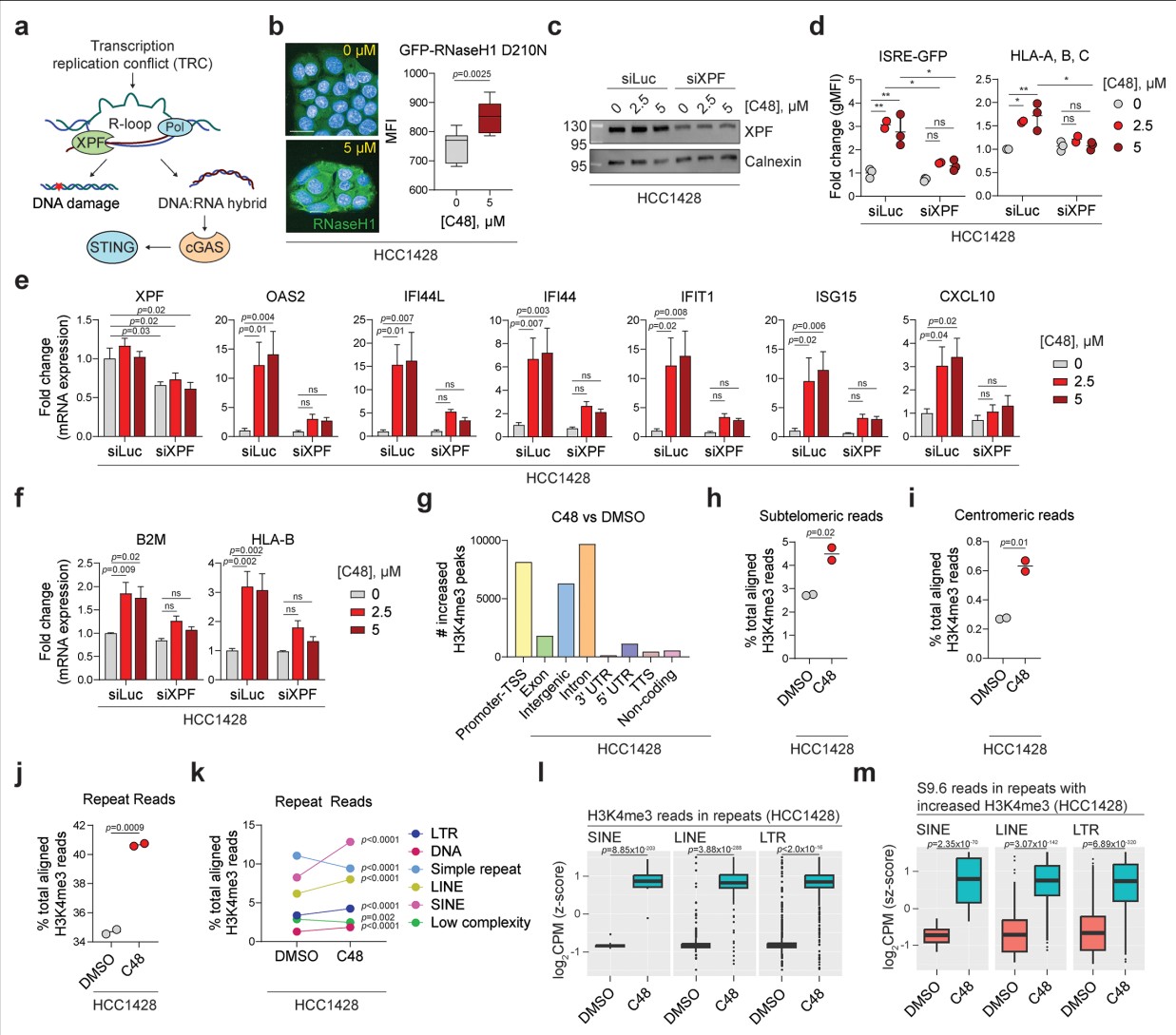

**Figure 4.** KDM5 inhibition induces R-loop formation in repeat regions that harbor increased H3K4me3. (**a**) Schematic of R-loops and their potential deleterious consequences. (**b**) Left: representative immunofluorescence images of cells incubated with GFP-RNaseH1 D210N protein (green), which detects RNA:DNA hybrids, in HCC1428 cells treated with mock and C48 as indicated. Blue = Hoechst. Scale bar = 50 μM. Right: mean fluorescence intensity (MFI) quantification of GFP positivity normalized to number of DAPI-positive nuclei. n=8 wells (two independent experiments where nine fields/ well were counted in triplicate), whiskers are from min to max, p-values are from unpaired *t*-test. (**c**) Representative immunoblot of XPF protein levels in HCC1428 cells transfected with siRNAs against luciferase (control) or XPF and treated as indicated. Samples were harvested 6 days post-transfection and after 5 days of treatment. Calnexin was used as a loading control. (**d**) FACS quantification of ISRE-GFP fluorescence and HLA-A, B, C surface levels in HCC1428 cells transfected and treated as in (**c**). Data are represented as fold change of geometric MFI (gMFI) over control (siLuciferase-transfected cells treated with DMSO, labeled as 0 μM C48). n=3, data are mean ± sem, p-values are from mixed-effects analysis with multiple comparisons. (**e**) qRT-PCR analysis of mRNA expression of XPF and genes representing ISG and (**f**) AP signatures, measured as 2^(Ct values of indicated gene subtracted from Ct value of β-actin), of the indicated genes in HCC1428 cells transfected and treated as in (**c**). Data are represented as fold change of mRNA expression levels over control. n=3–4, data are mean ± SEM, p-values are from mixed-effects analysis with multiple comparisons. ns = not significant. (**g**) Genomic annotations of spike-in-normalized H3K4me3 peaks that are significantly increased (log₂ fold change >0, p_adj<0.05) in C48-treated compared to DMSO-treated CUT-and-Tag generated data from HCC1428 cells. (**h**) Percent of total aligned H3K4me3 reads that map to sub-telomeric regions, as defined in *Stong et al., 2014* or (**i**) to centromeric regions, as defined in UCSC Genome Browser in H3K4me3 CUT-and-Tag data from cells treated as indicated. n=2, p-values are from unpaired *t*-tests. (**j**) Percent of total aligned H3K4me3 reads that map to repeat regions in CUT-and-Tag data from DMSO or C48-treated HCC1428 cells. n=2, p-value is from unpaired *t*-test. (**k**) as in (**j**), but separated into individual repeat classes. Only classes that represent over 3% of total aligned H3K4me3 reads are graphed. p-values are from Sidak's multiple comparisons test. (**l**) Log₂ counts per million (CPM) of H3K4me3 reads that map to the indicated class of repeats in CUT-and-Tag data from HCC1428 cells treated as indicated. Values are z-score scaled. Statistical test is a multiple regression model, and p-values are from the interaction terms of 'DMSO' and 'C48'. (**m**) Log₂CPM of S9.6 CUT-and-Tag generated reads

*Figure 4 continued on next page*

*Figure 4 continued*

in subsets of repeats that harbor significantly increased H3K4me3 after C48 treatment, separated into the indicated repeat classes. Values are z-score scaled. p-values are calculated as in (**I**).

The online version of this article includes the following source data and figure supplement(s) for figure 4:

**Source data 1.** Raw, unlabelled blot images corresponding to panel (c).

**Source data 2.** Raw blot images corresponding to panel (c) with target proteins labeled.

**Figure supplement 1.** KDM5 inhibition increases R-loop abundance in repeat regions that harbor increased H3K4me3.

experiments (*Kaya-Okur et al., 2019*) using an H3K4me3-specific antibody. This method combines antibody-targeted cleavage by Tn5 transposase with parallel DNA sequencing to evaluate regions in the genome that a target is bound. Since KDM5 inhibition induces global changes in H3K4me3, we added equal numbers of *Drosophila melanogaster* spike-in nuclei to both DMSO and C48-treated samples to enable downstream normalization (*Figure 4—figure supplement 1b* for experimental set-up as well as materials and methods for more detail). Antibodies against H3K4me3 and *Drosophila* spike-in were used to pull down human and *Drosophila* DNA, respectively, and libraries were generated for next-generation sequencing. Consistent with a global increase in H3K4me3, C48-treated samples consistently harbor less *Drosophila* reads compared to DMSO-treated samples (*Figure 4—figure supplement 1c*); therefore, H3K4me3 peaks would be undercounted in C48-treated samples without spike-in normalization (*Figure 4—figure supplement 1d and e* as examples). After peak counts were normalized to spike-in reads, differential peak calling was performed. As expected, we observe an increase in H3K4me3 in promoter regions upon C48 treatment in HCC1428 cells compared to DMSO controls (*Figure 4g*, *Figure 4—figure supplement 1e*, yellow bar). Genes that display increased H3K4me3 include ISGs; the modification of the genes may be directly mediated by KDM5, or they may be indirectly affected due to IFN pathway stimulation (*Figure 4—figure supplement 1f*). Additionally, we observe an increase in H3K4me3 in intergenic and intronic regions, which often harbor insertions of repetitive elements (*Elbarbary et al., 2016*), in C48-treated HCC1428 samples (*Figure 4g*, *Figure 4—figure supplement 1e*, blue and orange bars). Given that R-loop generation has been described in telomeric, centromeric, and other repeats (*Gong and Liu, 2023*; *Liu et al., 2021*), we expanded the analysis of H3K4 trimethylation to repeat regions. This analysis reveals increased H3K4me3 reads mapping to both sub-telomeric and centromeric regions following C48 treatment of HCC1428 and other luminal breast cancer cell lines (*Figure 4h and i*, *Figure 4—figure supplement 1g*). In this context, it is noteworthy that KDM5 has been identified as a telomere-associated protein (*Fujita et al., 2013*). The C48-mediated increases of H3K4me3 in sub-telomeric repeats occur in both tumor cells and normal HMECs, suggesting that repeats derived from these regions likely do not contribute to the 'viral mimicry" phenotypes observed (data not shown).

In addition to the observed C48-mediated increases of H3K4me3 in telomeric repeats, after subdividing the genome into annotated repetitive regions and non-repeat regions, we also detect a significant increase in H3K4me3 reads mapping to repeat regions in HCC1428 cells (*Figure 4j*) and other breast tumor cell lines (*Figure 4—figure supplement 1h*). The results of the H3K4 me3-CUT-and-Tag experiments are consistent with those derived from the analysis of H3K4me3 reads from chromatin immunoprecipitation (ChIPseq) experiments (data not shown). We subsequently separated the analysis by repeat classes present in the human genome and observed that SINE elements exhibit the highest coverage of H3K4me3 in C48-treated HCC1428 cells (*Figure 4—figure supplement 1i*, pink bar) and are significantly increased in comparison to untreated samples (*Figure 4k*, pink). We also identify increases in H3K4me3 in other genomic repeats such as LINE and LTR elements in HCC1428 cells treated with C48 (*Figure 4k*, yellow and dark blue respectively, *Figure 4l*) and in other C48-treated luminal breast cancer cell lines (*Figure 4—figure supplement 1h* and data not shown). Further distillation of repeat classes into SINE, LINE, and LTR subfamilies reveals that KDM5 inhibition results in increased H3K4me3 mainly in the AluY subfamily, as well as LINE1/LINE2 subfamilies and a few LTR subfamilies (*Figure 4—figure supplement 1j and k*).

To investigate a possible relationship between genomic regions that harbor increased H3K4me3 upon KDM5 inhibition and R-loop prevalence, we next performed CUT-and-Tag experiments using an antibody (S9.6) that detects RNA:DNA hybrids in the genome. As in the H3K4me3-CUT-and-Tag experiments, *Drosophila* 'spike-ins' were added and used for normalization of the data (*Figure 4—figure*

*supplement 1b* for experimental outline and 'Materials and methods' for more detail). Similar to what was observed for H3K4me3, KDM5 inhibition increases S9.6 reads mapping to telomeric and centromeric repeats in HCC1428 cells (*Figure 4—figure supplement 1l*). Since we also observed H3K4me3 increases mapping to other repeat classes following KDM5 inhibitor exposure, we expanded the S9.6 CUT-and-Tag analysis to the rest of the repeat genome. This analysis shows that repeats that harbor KDM5 inhibitor-mediated increases in both H3K4me3 and S9.6 reads traverse multiple repeat classes, including SINEs, LINEs, and LTRs (*Figure 4m*). Repeats that do not display increases in H3K4me3 also do not show a significant increase in S9.6 reads (*Figure 4—figure supplement 1o*). Formation of R-loops can be linked to increased transcription in repeat regions; therefore, we separated repeat elements further into three groups: those that harbor increases in H3K4me3, those that have increased mRNA expression (as measured by RNA sequencing), and those that display both (*Figure 4—figure supplement 1m*). While repeats in all three groups display increases in S9.6 reads upon C48 treatment (*Figure 4—figure supplement 1n and o*), this increase is greatest in repeats that harbor increases in both H3K4me3 and transcription (*Figure 4—figure supplement 1n*, middle panel). Together, these analyses demonstrate that KDM5 inhibition and subsequent increases in H3K4me3 and transcription most strongly correlate with an increased prevalence of R-loops in genomic repeats in tumor cells.

## Discussion

Collectively, we show that the KDM5 demethylases regulate 'viral mimicry" and DNA damage in luminal breast cancer cells. Mechanistically, KDM5 inhibition increases R-loop prevalence in repetitive genomic regions, triggering activation of the cGAS/STING pathway and consequent upregulation of ISGs and MHC class I surface expression. Such increases may result in combinatorial effects with immunotherapy drugs. KDM5 inhibitors also reduce cell fitness, not mediated through induction of IFN-I signaling, but rather because of a DNA damage response generated as a consequence of increased R-loop formation (*Figure 5* for model). As mentioned above, R-loops can form during transcription and have the potential to impede replication fork progression, leading to replication stress. This interference can result in DNA damage, necessitating repair mechanisms to maintain genomic integrity. Conversely, replication stress itself might induce conditions conducive to R-loop formation, suggesting a bidirectional relationship. The study of chromatin dynamics and its impact on DNA damage and repair mechanisms has garnered significant attention in recent years. Several studies have implicated KDM5 demethylases in DDR (*Gaillard et al., 2021*; *Kumbhar et al., 2021*), and a recent study revealed a potential role for KDM5 in maintaining the delicate balance between transcriptional regulation and replication to preserve DNA integrity (*Wang et al., 2023*). Collectively, such

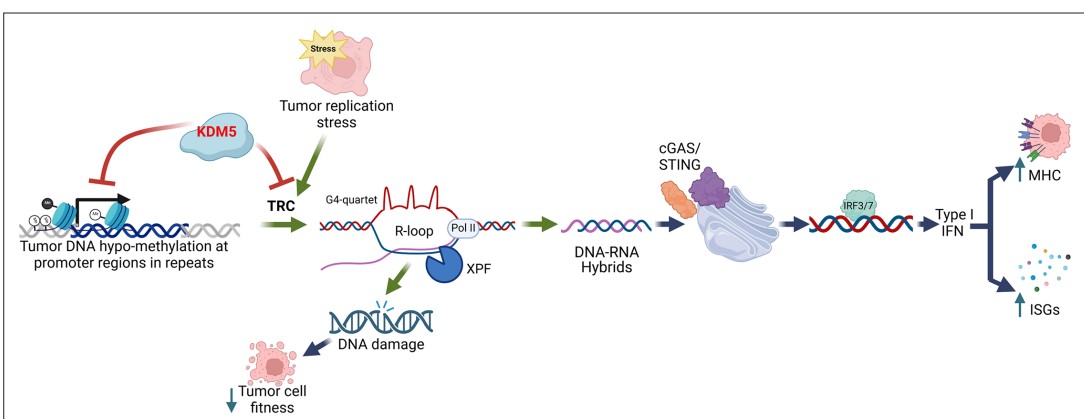

**Figure 5.** The KDM5 family of epigenetic regulatory enzymes reduces R-loop formation in genomic repeat regions in luminal breast cancer cells. Genetic disruption or chemical inhibition of KDM5 causes R-loop-mediated DNA damage in luminal breast cancer cells. R-loop generated RNA:DNA hybrids also result in the activation of the cGAS/STING pathway, increasing ISG and AP signatures, as well as cell surface MHCI, which may enhance interactions with the TME. Importantly, KDM5 inhibition does not result in DNA damage or activation of the cGAS/STING pathway in normal breast epithelial cells, suggesting that KDM5 inhibitors present a wider therapeutic window in this setting than STING agonists or type I interferons. Created in BioRender.

observations suggest that KDM5 inhibitors, or other factors whose disruption or inhibition results in tumor-specific increases in R-loops, could also pair well with PARP inhibitors and other DNA damage-promoting drugs.

Importantly, we also show that KDM5 inhibition selectively affects cell fitness and immunogenicity phenotypes of cancer cells without having similar effects in primary breast epithelial cells, potentially resulting in a therapeutic window for KDM5 inhibitors or degraders as single agents. Our data also show that KDM5 disruption or inhibition can activate T cells in a co-culture system. Important for possible combinations with TME enhancers, preliminary data suggest that there are no major negative effects of KDM5 inhibitors on primary T cells or macrophages; in fact, KDM5 inhibition may induce a more anti-tumorigenic phenotype (*Figure 3—figure supplement 2* and data not shown). The STING-independent KDM5 inhibitor effect on cell fitness in luminal breast cancer cells and previous observations that KDM5 disruption affects the survival of drug-tolerant persister cells (*Sharma et al., 2010*; *Vinogradova et al., 2016*) suggest that there could be combinatorial potential, not only with agents that affect the TME, but also with other standard-of-care agents. Future studies will also have to explore the relevance of these findings in other cancer subtypes. For example, preliminary studies suggest that hematologic cancer cells that are sensitive to KDM5 inhibitors also display markers of IFN-I signaling activation (data not shown). Collectively, these findings expand on KDM5 inhibitors or degraders as promising candidates for single-agent cancer therapy, with further potential to enhance the efficacy of immunotherapies as well as DNA damaging agents.

# Materials and methods
## Cell lines
HCC1428, SKBR3, and MCF7 were purchased from ATCC (CRL-2327, HTB-30, and HTB-22, respectively). Jurkat expressing a luciferase reporter construct driven by an NFAT-response element was purchased from Promega (J1621). HMECs were purchased from Lonza (CC-2551). All cell lines were regularly checked for mycoplasm.

## Cell culture conditions
HCC1428, SKBR3, MCF7, and Jurkat cells were maintained at 37°C with 5% $CO_2$ RPMI1640 containing 10% heat-inactivated fetal bovine serum (FBS, Sigma) and 4.5 g/l glucose. HMEC cells were cultured in MEBM Basal Medium supplemented with MEGM SingleQuots (Lonza CC-3150). Experiments using HMEC cells were completed before 15 passages.

## Inhibitors/agonists used
KDM5 inhibitor C48 was synthesized at Pfizer. Recombinant IFN-β was purchased from PBL Assay Sciences (cat 11410-2). STING agonist di-ABZI was purchased from Selleck Chemicals (cat S8796). JAK inhibitor ruxolitinib (cat tlrl-rux) and low molecular weight polyIC (cat tlrl-picw) were purchased from InvivoGen. IR-Alu was synthesized by reverse transcription.

## Cell line generation
### ISRE-GFP containing cell lines
Lentivirus containing ISRE-GFP was purchased from GenTarget (cat LVP937-N). Cells were seeded to 90–100% confluence in 6-well plates. Lentiviral particles were combined with 2 ml media at ~15–20% MOI. Viral media was removed the day after and replaced with fresh media. Selection in media supplemented with 200 ug/ml geneticin (Gibco) was performed the day after, for 7 days. Cell line pools were used for experiments. NY-ESO1: MCF7 cells were transduced with lentiviral particles expressing NY-ESO1, followed by stable selection in the presence of 1 ug/ml puromycin. NY-ESO1 TCR/CD3: Jurkat cells expressing NFAT-response element-driven luciferase reporter were cultured as described above supplemented with 200 ug/ml hygromycin B. Cells were transduced with a lentiviral construct expressing NY-ESO1 TCR and selected with 2 ug/ml puromycin to generate a stable pool of Jurkat NY-ESO1 TCR/CD3 cells. Pools were used for experiments that were run with no selection media.

## Immunoblotting

Cells were lysed in RIPA buffer (50 mM Tris HCl, pH 7.4, 150 mM NaCl, 0.5% deoxycholate, 0.1% sodium dodecyl sulfate, 1% NP-40; Thermo 89900) with proteinase/phosphatase inhibitor cocktail (Thermo) and benzonase (Sigma). Whole-cell extracts were sonicated via Active Motif Pixul multi-sample sonicator, then assayed for protein concentration via BCA (Thermo). Samples were leveled and denatured for 10 minutes at 70°C, then mixed with 4X LDS sample buffer (Thermo NP0008) and 10X sample reducing agent (Thermo NP0009). Samples were loaded at 1 µg for H3 and H3K4me3, and at 20 µg for all other proteins, then separated by electrophoresis on a 4–20% precast polyacrylamide gel (Bio-Rad 4561096), transferred using Bio-Rad TransBlot Turbo to nitrocellulose membranes, and blocked in blocking buffer (Rockland NC0168431) for 20 minutes. Membranes were incubated with primary antibodies overnight at 4°C on a rocking platform, followed by washing in 0.1% Tween/PBS. Membranes were incubated with appropriate HRP-linked secondary antibodies at room temperature (RT) for 1 hour and washed 3× before signal detection. Membranes were developed by chemiluminescence using Bio-Rad ECL reagent (cat 1705062). Images were acquired with the Bio-Rad ChemiDoc imaging system. Primary antibodies used were ADAR (Invitrogen MAB-31609), Actin (Millipore A5541), H3K4me3 (Cell Signaling 9751), H3 (Cell Signaling 4499), STING (Cell Signaling 13647), Tubulin (Cell Signaling 2128), cGAS (Cell Signaling 15102, 79978), KDM5A (Bethyl Labs A300-897A), KDM5B (Bethyl Labs A301-813A), KDM5C (Bethyl Labs A301-034A), XPF (Cell Signaling 13465), and Calnexin (Cell Signaling 2679). Secondary antibodies used were anti-mouse IgG, HRP linked and anti-rabbit IgG (Cell Signaling 7076), and HRP linked (Cell Signaling 7074).

## Immunofluorescence

Cells were plated in 96-well black Cell Carrier plates (PerkinElmer 6055302) 2–3 days before harvest. On the day of harvest, plates were washed with PBS and fixed in ice-cold methanol at –20°C for 10 minutes. After 3× PBS wash, cells were permeabilized with 0.24% Triton X-100 in PBS, 3 minutes on ice. Cells were then incubated in 50 µl of 1x RNaseH buffer supplemented with 20 mM MgCl$_2$, with or without RNaseH diluted 1:50 (NEB M0297), for 3 hours at 37°C. After 3× PBS wash, cells were incubated in blocking solution (3% BSA in PBS) for 1 hour at RT. Cells were then incubated with GFP-RNaseH1::D210N protein at 0.05 µg/ml or anti-γH2AX antibody (Cell Signaling 2577) at 1:500, diluted in blocking solution, overnight at 4°C. The day after, the plate was incubated at 37°C for an additional 1.5 hours to intensify signal. Cells were washed 3× in PBS, 5 minute rocking, then incubated in secondary antibody at 1:1000 to probe γH2AX (AlexaFluor 488 anti-rabbit IgG, Thermo A-21206) and Hoescht (Invitrogen H3570) at 1:2000, diluted in blocking solution, for 1 hour at RT. After final three washes in PBS (5 minute each rocking), 100 µl PBS was left in wells for image acquisition.

## Image acquisition and analysis

Images were acquired on a PerkinElmer Operetta CLS equipped with 63× water immersion objective, using Two Peak autofocus and confocal optic mode. Image analysis was performed using Harmony (version 4.9) software and exported with equal contrast adjustment. The HOECHST 33258 channel was used to identify nuclei using the Find Nuclei module in the analysis sequence, using method C with a common threshold of 0.40, area of 30 µm, splitting coefficient of 9.8, individual threshold of 0.30, and contrast of 0. The EGFP channel was used to identify the cytoplasm region using the Find Cytoplasm module in the analysis sequence, using the nuclei identified in the previous step, via method A, and an individual threshold of 0.15. Cell regions of Cell and Cytoplasm were output in this step. Mean fluorescence intensities (MFI) for GFP-dRNaseH1 and γH2AX were calculated for each subcellular region and channel of interest using the Calculate Intensity Properties module and were exported along with nucleus counts using the Define Results module. MFI of RNaseH1 and γH2AX was normalized to number of DAPI-positive nuclei. For each experiment, nine fields were analyzed in triplicate for two independent experiments.

## Cellular fitness/growth curve analysis

Cells were plated into 24-well, 12-well, or 6-well plates and treated as indicated. For culture over 7 days, cells were split on day 7 of the experiment. All cells were split at the same ratios regardless of confluence. Whole-well phase-only images were acquired on IncuCyte (Sartorius) at the indicated times. Confluence was calculated using IncuCyte software after setting the following parameters:

segmentation adjustment = 1.5, area min = 600 μm², eccentricity max = 0.990. To calculate doubling times for statistical analysis, confluence for each day was normalized to that of day 0 in each sample. $Log_2$ of each normalized confluence timepoint was calculated to generate a linear curve (excluding day 0 timepoint), and slope was calculated from each line. Doubling time was calculated as 1/slope of each line.

## Growth inhibition assays

Assays were performed by ChemPartner Co., Ltd (Shanghai, China). Briefly, breast cancer cell lines were treated with varying concentrations of C48, and viability was determined via CyQUANT assays, $IC_{50}$ values were calculated, and inhibition was calculated as area under the $IC_{50}$ curve (AUC) after 21 days of treatment.

## Cell assays/drug treatments

For KDM5 inhibitor experiments, cells were treated with KDM5-C48 at the indicated concentrations for 7 days (unless otherwise noted). In experiments using siXPF, cells were treated with C48 at the indicated concentrations 1 day after transfection for a total of 5 days. In JAK inhibition experiments, cells were treated with 1 μM ruxolitinib for the duration of the experiment. Media and drugs were changed every 2–3 days. For agonist experiments, cells were treated with recombinant IFN-β at 1000 U/ml, di-ABZI at 1 μM, or transfected with 2.5 μg/ml low molecular weight polyIC or 300 ng/ml IR-Alu using RNAiMAX for the last 24 hours.

## siRNA knock-down

For individual knockdown, HCC1428 cells in 6-well plates were transfected at 25 nM final concentration with 5 μl RNAiMAX (Thermo) per well in OptiMEM. siRNAs used (*Crossley et al., 2023*) were

> siLuciferase (siLuc): 5'-CUUACGCUGAGUACUUCGA-3'
> siXPF: 5'-ACAAGACAAUCCGCCAUUA-3'

## CRISPR-mediated gene disruption

crRNAs (ordered from IDT) were combined with tracrRNA (IDT 1072533) at equal concentration (200 μM each), heated to 95°C for 5 minutes and allowed to cool to RT to anneal. 0.3 μl Cas9 protein (IDT 1081059) was combined with 0.2 μl buffer R from Neon Transfection System 10 μl Kit (Thermo MPK1096); 0.5 μl of the Cas9 mix was then combined with 0.5 μl of annealed RNA duplex or sgRNA (IDT) for 10–20 minutes at RT. 2 μl of Alt-R Electroporation Enhancer, 10.8 μM stock concentration (IDT 1075916) was added to the mixture, followed by 9 ul of $1.2 \times 10^5$ cells resuspended in buffer R. 10 μl of cell/sgRNA/Cas9 mixture (approximately $1 \times 10^5$ cells) was then electroporated using Neon Transfection System (Thermo), setting #3, and plated into 6-well plates. For electroporation of $1 \times 10^6$ million cells, all components were multiplied by 10 and Neon Transfection System 100 μl Kit (Thermo MPK10096) was used, and cells were plated in 10 cm plates. For CRISPR disruption of ADAR, cells were harvested for immunoblot 4 days post-electroporation, and for flow cytometry and RNA isolation 7 days post-electroporation. For KDM5 paralog CRISPR disruptions, cells were harvested 7 days post-electroporation for downstream analysis. For cell growth analyses via IncuCyte, the day of electroporation was designated day 0. For cGAS/STING CRISPR-mediated disruptions, cells were cultured for 3 days before KDM5 inhibition. Individual Cas9-mediated gene disruptions were performed for each experimental replicate. crRNA/sgRNA sequences are the following:

crRNA:

> sgHPRT1i (intronic cutting control): 5'-AATTATGGGGATTACTAGGA-3'
> sgADAR #3: 5'-ATACCTGAACACCAACCCTG-3'
> sgADAR #4: 5'-AGCCGAATGCCATCCCACGT-3'
> sgTMEM173 #1: 5'-GGCAAACAAAGTCTGCAAGG-3'
> sgTMEM173 #3: 5'-CCTCACCCTGGTAGGCAATG-3'
> sgMB21D1 #2: 5'-TAATAAGAAGTGTTACAGCA-3'
> sgMB21D1 #3: 5'-GAGCTACTATGAGCACGTGA-3'

sgRNA:

sgKDM5Ai (intronic cutting control): 5'-GAGGTAATGGAGTACCTAAG-3'
sgKDM5A #1: 5'-ATCTACTCTGAAGATCCCTG-3'
sgKDM5A #4: 5'-TCTGTGAACTCCTCCCAACT-3'
sgKDM5Bi (intronic cutting control): 5'-AGGACTGGGCGATTAGACTG-3'
sgKDM5B #1: 5'-AACCCTGGTTAAAACCACTG-3'
sgKDM5B #5: 5'-AACCCCATGATATTCCCCAG-3'
sgKDM5Ci (intronic cutting control): 5'-AGGACTGGGCGATTAGACTG-3'
sgKDM5C #3: 5'-CGCCTTGAGCGCATACACAG-3'
sgKDM5C #4: 5'-TGGCACACCATGGGACATGA-3'

## RNA isolation and qRT-PCR

Cells were harvested on the indicated days and total RNA was isolated using Qiagen RNeasy kit following manufacturer's protocol and subjected to DNase treatment. For qRT-PCR analysis, 10 ng of RNA was subjected to one-step cDNA synthesis and amplification using 1x Taqman Fast Virus 1-Step Master Mix (Applied Biosystems 4444436), along with 1x Taqman probes. Samples were run in duplicate in 384-well plates (Applied Biosystems 4483321) on ViiA7 (Applied Biosystems) using the following conditions: 48°C for 15 minutes (cDNA synthesis), 95°C for 20 seconds (inactivation/denaturation), followed by 95°C for 3 seconds (denaturation) and 60°C for 1 minute (elongation) for a total of 40 cycles. Delta Ct was calculated from subtracting Ct values of gene of interest from B-actin Ct values. Raw values are expressed as 2^delta Ct. Probes used were purchased from Thermo and are as follows: B-actin (Hs01060665_g1), OAS2 (Hs04185073_m1), IFI44L (Hs0019915_m1), IFI44 (Hs00951349_m1), ISG15 (Hs00192713_m1), IFIT1 (Hs01911452_s1), TMEM173 (Hs00736955_g1), MB21D1 (Hs00403553_m1), IFNL1 (Hs00601677_g1), CXCL10 (Hs00171042_m1), B2M (Hs00187842_m1), HLA-B (Hs00818803_g1), ERCC4 (Hs01063530_m1).

## RNASeq sample generation and analysis

Cells were treated with DMSO or 2.5 µM C48 for 2 or 7 days. RNA was prepared as described above. Samples were generated in triplicate. RNA was randomly primed (not poly(A)-selected), rRNA depleted, and sequenced via Illumina NovaSeq using 150 bp paired-end read sequencing with a sequencing depth of 100 million reads (by Azenta Life Sciences). RNAseq fastq files were aligned to the human GENCODE v28 transcriptome using STAR aligner (v2.6.0c) and gene expression was quantified using RSEM (v.1.3.0). Data4Cure was utilized for analysis (https://www.data4cure.com). ISG and AP gene lists were generated as described (see 'Gene set variation analysis').

## CUT-and-Tag sample generation and analysis

Cells were treated with DMSO or 2.5 µM C48 for 5 days. Samples were generated in duplicate, cryo-preserved, and sent to Active Motif for downstream preparation. Cryopreserved cells were thawed from –80°C in a 37°C water bath with gentle movement until no ice was visible. Cell samples were then gently pipetted up and down to resuspend cells, and all liquid was transferred to 2 ml round bottom snap-cap microcentrifuge tubes (Fisher 14-666-315). Samples were centrifuged at 600 × G in a centrifuge at 4°C for 3 minutes. Liquid was carefully removed with a pipettor, taking care to avoid cell pellets. Samples were then resuspended in DIG-Wash buffer with Proteinase Inhibitor cocktail and processed according to the CUT&Tag-IT Assay Kit, Anti-Rabbit (Active Motif 53160) protocol for H3K4me3 CUT&Tag samples and CUT&Tag-IT R-loop Assay Kit (Active Motif 53167) for R-loop samples, with the addition of *Drosophila* spike-in nuclei and spike-in antibody as follows: *Drosophila* spike-in nuclei CUT&Tag-IT Spike-In Control, Anti-Rabbit (Active Motif 53168) for H3K4me3 CUT&Tag and CUT&Tag-IT Spike-In Control, R-loop (Active Motif 53174) was thawed from –80°C in a 37°C water bath with gentle movement until no ice was visible. Spike-in nuclei were gently pipetted up and down to resuspend and homogenize the suspension. Spike-in nuclei were added to each sample at a ratio of 1:10 spike-in nuclei to cells, and each was added to ConA beads. Cells, nuclei, and ConA beads were mixed by gentle pipetting, and strip tubes containing the samples were placed on a room temperature nutator for 10 minutes. Spike-in antibody was added to each sample according to the kit instructions at the same time as the primary antibody, H3K4me3 (Active Motif 39160) or S9.6 (included in R-loop CUT&Tag kit) was added to the cell sample/spike-in nuclei mix. The CUT&Tag-IT R-loop Assay Kit protocol was followed as per the manufacturer's instructions for all remaining steps for

R-loop samples and CUT&Tag-IT Assay Kit, Anti-Rabbit for the H3K4me3 samples. NGS libraries were evaluated on a TapeStation 4150 (Agilent, Santa Clara, CA) using High Sensitivity D1000 ScreenTapes (Agilent 5067-5584) and pooled for equal sequencing depths targeting approximately 30 million reads per sample. Sequencing was conducted on a NovaSeq6000 (Illumina, San Diego, CA).

Samples were aligned to hg38 and dm6 genome using Bowtie2 (v2.3.4.1). Samples were normalized according to Active Motif's normalization protocol. Briefly, uniquely aligning *Drosophila* tags pulled down by H2Av were counted. Normalization factor was generated by dividing *Drosophila* counts of sample with lowest tag count by the *Drosophila* count of each sample. To find the location of peaks in the genome, peak positions were first called per sample using Homer (http://homer.ucsd.edu/homer/), then merged into a set of consensus genomic positions. They were then annotated, and peak scores were calculated per sample. To count reads aligning to repetitive elements, RepEnrich2 was utilized (**Skvir and Criscione, 2022**) Repeat reference was downloaded from UCSC Genome Browser (hg38; https://genome.ucsc.edu/). To calculate percent reads mapping to repeat regions, the sum of total counts output by RepEnrich2 was divided by the number of reads that align concurrently at least once. Differential analysis was performed using edgeR, using normalization factors as the library size. Counts per million (CPM) was also calculated using normalization factors as the library size. To calculate the percentage of reads in subtelomeric regions, samples were aligned to subtelomeric sequences (**Stong et al., 2014**) using Bowtie2. Centromeric sequences were downloaded from UCSC Genome Browser, and Bowtie2 was used to determine percentage of reads that align to centromeric regions.

## GEO numbers
GEO accession numbers are as follows:

GSE296387: H3K4me3 CUT-and-Tag data
GSE296584: S9.6 CUT-and-Tag data
GSE296974: RNA-sequencing data

## Gene set variation analysis
To obtain enrichment scores, gene set variation analysis (GSVA) was performed using the GSVA R package (**Hänzelmann et al., 2013**) version 3.16 with Gaussian kernels. IFN/AP gene sets are provided in **Supplementary file 1**. The DepMap 22Q2 CCLE dataset (https://depmap.org/portal/) was utilized as the gene expression data. GSVA scores were plotted by cancer type, as well as breast cancer subtype as defined by **Dai et al., 2017**.

## Gene dependency
Gene dependency (CERES) scores were obtained from DepMap, 22Q2 (https://depmap.org/portal/).

## Co-culture experiments
The day before co-culture, 40,000 MCF7::NY-ESO1 cells were plated in white 96-well plates (Corning 3917) in triplicate. The next day, 100,000 Jurkat::NY-ESO1 TCR cells were plated in each well in 75 µl fresh media supplemented with 25 mM HEPES (Gibco 15630-080). After 5 hours of co-culture, the plate was taken out of the incubator and rested at room temperature for 5 minutes. 75 µl of BioGlo Luciferase Reagent (Promega G7940) was added to each well and incubated for 5-minute shaking before reading on EnVision 2104 (PerkinElmer) on default luminescence settings.

## Flow cytometry
Cells (at least 100,000) were dissociated using accutase (Gibco A1110501) and transferred to 96-well 1 ml plates (Greiner Bio-One 780261). After spinning down 5 minutes at 400 × *g*, cells were incubated in 50 µl FACS buffer (1× PBS + 2% FBS) containing antibody (5 µl) and live/dead stain (1 µl of 1:5 dilution, Thermo L34975) for 30 minutes at 4°C. Cells were then washed with 1 ml FACS buffer, centrifuged as above, and resuspended in 100 µl FACS buffer. Cells were then moved to 96-well plates (Falcon 353263) and run through LSR Fortessa (BD Biosciences). Antibodies used are the following: AlexaFluor 488 anti-human HLA-A, B, C clone W6/32 (BioLegend 311413), and APC anti-human HLA-A, B, C clone W6/32 (BioLegend 311410).

### KDM5 paralog/ADAR dependency in mouse

Data was extracted from the Genome data provided in https://tumorimmunity.org/#/ (*Dubrot et al., 2022*), comparing sgKDM5 and sgADAR dependency between immune-competent (ICB) mice and immune-incompetent (NSG) mice.

### Statistical analysis

For all experiments, p-values are calculated as indicated in figure legends. Symbols are as follows: n.s. >0.05, *<0.05, **<0.1, ***<0.001, and ****<0.0001.

## Acknowledgements

We would like to thank Thomas Paul, Fred Derheimer, Jon Oyer, David Shields, Scott Tria, Michelle Wagner, William Snyder, and many other colleagues from the Pfizer CTI and Oncology organizations for discussions throughout the course of this work. We also thank Jeff Settleman for reading the manuscript.

## Additional information

### Competing interests

Lena Lau, Kurt Henderson, Ahu Turkoz, Anders Mälarstig, Chames Kermi, Paul Moore: affiliated with Pfizer at the time of the experiments. No other competing interests to declare. Sara Linker, Robert Rollins, Oleg Brodsky, Clifford Restaino, Kristen Jensen-Pergakes: employee of Pfizer and may hold stock in the company. No other competing interests to declare. Brad Townsley, Brian Egan: employee of Active Motif. No other competing interests to declare. Murali Gururajan, Marie Classon: affiliated with Pfizer at the time of the experiments and may hold stock in the company. No other competing interests to declare. The other authors declare that no competing interests exist.

### Funding

No external funding was received for this work.

### Author contributions

Lena Lau, Data curation, Formal analysis, Investigation, Methodology, Writing – original draft, Writing – review and editing; Kurt Henderson, Data curation, Formal analysis, Investigation, Methodology; Ahu Turkoz, Data curation, Formal analysis, Methodology; Sara Linker, Data curation, Methodology, Writing – original draft; Dorte Schlessinger, Data curation; Brad Townsley, Brian Egan, Conceptualization, Methodology; Shoba Ragunathan, Xianju Bi, Zhijian J Chen, Oleg Brodsky, Methodology; Robert Rollins, Supervision, Methodology; Clifford Restaino, Data curation, Investigation; Murali Gururajan, Investigation; Kristen Jensen-Pergakes, Supervision, Project administration; Anders Mälarstig, Data curation, Supervision; Chames Kermi, Conceptualization, Data curation, Supervision, Investigation; Paul Moore, Conceptualization, Data curation, Supervision, Investigation, Methodology, Writing – original draft; Marie Classon, Conceptualization, Supervision, Writing – original draft, Writing – review and editing

### Author ORCIDs

Lena Lau ⓘ https://orcid.org/0009-0001-8110-4949
Sara Linker ⓘ https://orcid.org/0000-0002-6653-1715
Brian Egan ⓘ https://orcid.org/0000-0002-6354-6575
Zhijian J Chen ⓘ https://orcid.org/0000-0002-8475-8251
Paul Moore ⓘ https://orcid.org/0009-0007-0460-0643
Marie Classon ⓘ https://orcid.org/0009-0007-3305-1890

Reviewer #1 (Public review): https://doi.org/10.7554/eLife.106249.3.sa1
Reviewer #2 (Public review): https://doi.org/10.7554/eLife.106249.3.sa2
Author response https://doi.org/10.7554/eLife.106249.3.sa3

# Additional files

## Supplementary files

MDAR checklist

Supplementary file 1. List of genes included in IFN-1 and AP signatures.

## Data availability

Sequencing data has been deposited in GEO under the accession codes: GSE296387: H3K4me3 CUT-and-Tag data; GSE296584: S9.6 CUT-and-Tag data; GSE296974: RNA-sequencing data; GSVA Analysis.

The following datasets were generated:

| Author(s) | Year | Dataset title | Dataset URL | Database and Identifier |
|---|---|---|---|---|
| Lau L, Henderson K, Turkoz A, Linker S, Schlesinger D, Townsley B, Egan B, Regunathan S, Rollins R, Bi X, Chen Z, Brodsky O, Restaino C, Gururajan M, Jensen-Pergakes K, Malarstig A, Kermi C, Moore P, Classon M | 2025 | KDM5 demethylases suppress R-loop-mediated "viral mimicry" and DNA damage in breast cancer cells [CUT&Tag_H3K4me3] | https://www.ncbi.nlm.nih.gov/geo/query/acc.cgi?acc=GSE296387 | NCBI Gene Expression Omnibus, GSE296387 |
| Lau L, Henderson K, Turkoz A, Linker S, Schlesinger D, Townsley B, Egan B, Regunathan S, Rollins R, Bi X, Chen Z, Brodsky O, Restaino C, Gururajan M, Jensen-Pergakes K, Malarstig A, Kermi C, Moore P, Classon M | 2025 | KDM5 demethylases suppress R-loop-mediated "viral mimicry" and DNA damage in breast cancer cells [CUT&Tag_S96] | https://www.ncbi.nlm.nih.gov/geo/query/acc.cgi?acc=GSE296584 | NCBI Gene Expression Omnibus, GSE296584 |
| Lau L, Henderson K, Turkoz A, Linker S, Schlesinger D, Townsley B, Egan B, Regunathan S, Rollins R, Bi X, Chen Z, Brodsky O, Restaino C, Gururajan M, Jensen-Pergakes K, Malarstig A, Kermi C, Moore P, Classon M | 2025 | KDM5 demethylases suppress R-loop-mediated "viral mimicry" and DNA damage in breast cancer cells [RNA-Seq] | https://www.ncbi.nlm.nih.gov/geo/query/acc.cgi?acc=GSE296974 | NCBI Gene Expression Omnibus, GSE296974 |

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
