## [Editor Report · eLife Assessment]

This study presents a **valuable** finding that KDM5 inhibitors may enable a wide therapeutic window as compared to STING agonists or type I interferons. The evidence supporting the claims of the authors is **convincing**. The work will be of broad interest to scientists working in the field of breast cancer research.

---

## [Referee Report · Reviewer #1 (Public review)]

In this manuscript, Lau et al reported that KDM5 inhibition in luminal breast cancer cells results in R-loop-mediated DNA damage, reduced cell fitness and an increase in ISG and AP signatures as well as cell surface Major Histocompatibility Complex (MHC) class I, mediated by RNA:DNA hybrid activation of the CGAS/STING pathway.

Their studies have shown that KDM5 inhibition/loss mediates a viral mimicry and DNA damage response through the generation of R-loops in genomic repeats. This is a different mechanism from the more well studied double-stranded RNA-induced "viral mimicry" response.

More importantly, they have shown that KDM5 inhibition does not result in DNA damage or activation of the CGAS/STING pathway in normal breast epithelial cells, suggesting that KDM5 inhibitors may enable a wide therapeutic window in this setting, as compared to STING agonists or Type I Interferons.

Their findings provide new insights into the interplay between epigenetic regulation of genomic repeats, R-loop formation, innate immunity, and cell fitness in the context of cancer evolution and therapeutic vulnerability.

Comments on revised version:

The authors have satisfactorily addressed my comments and revised the manuscript accordingly.

---

## [Referee Report · Reviewer #2 (Public review)]

Summary:

In this manuscript, the authors investigated how the type-I interferon response (ISG) and antigen presentation (AP) pathways are repressed in luminal breast cancer cells and how this repression can be overcome. They found that a STING agonist can reactivate these pathways in breast cancer cells, but it also does so in normal cells, suggesting that this is not a good way to create a therapeutic window. Depletion of ADAR and inhibition of KDM5 also activate ISG and AP genes. The activation of ISG and AP genes is dependent on cGAS/STING and the JAK kinase. Interestingly, although both ADAR depletion and KDM5 inhibition activate ISG and AP genes, their effects on cell fitness are different. Furthermore, KDM5 inhibitor selectively activates ISG and AP genes in tumor cells but not normal cells, arguing that it may create a larger therapeutic window than the STING agonist. These results also suggest that KDM5 inhibition may activate ISG and AP genes in a way different from ADAR loss, and this process may affect tumor cell fitness independently of the activation of ISG and AP genes.

The authors further showed that KDM5 inhibition increases R-loops and DNA damage in tumor cells, and XPF, a nuclease that cuts R-loops, is required for the activation of ISG and AP genes. Using H3K4me3 CUT&RUN, they found that KMD5 inhibition results in increased H3K4me3 not only at genes, but also at repetitive elements including SINE, LINE, LTR, telomeres, and centromeres. Using S9.6 CUT&TAG, they confirmed that R-loops are increased at SINE, LINE, and LTR repeated with increased H3K4me3. Together, the results of this study suggest that KMD5 inhibition leads to H3K4me3 and R-loop accumulation in repetitive elements, which induces DNA damage and cGAS/STING activation and subsequently activates AP genes. This provides an exciting approach to stimulate the anti-tumor immunity against breast tumors.

KDM5 inhibition activates interferon and antigen presentation genes through R-loops.

Strengths:

A new approach to make breast tumors "hot" for anti-tumor immunity.

Weaknesses:

Future in vivo studies are needed to show the effects of KDM5 inhibitors on the immunotherapy responses of breast tumors.

Comments on revised version:

The authors have adequately addressed my comments.

---

## [Author Response]

The following is the authors’ response to the original reviews

We thank the reviewers for their careful and positive assessment of our manuscript. Maybe our findings are best summarized in the model below, showing that KDM5 inhibition/loss mediates a viral mimicry and DNA damage response through the generation of R-loops in genomic repeats. This is a different mechanism from the more well studied double-stranded RNA-induced “viral mimicry” response. Our studies also suggest that KDM5 inhibition may have a larger therapeutic window than STING agonists, since KDM5 inhibition seemingly does not induce “viral mimicry” in normal breast epithelial cells.

**Author response image 1. sa3fig1:** Model of viral mimicry activation. De-repression of repetitive elements may trigger dsRNA formation, which activates the RIG-1/MDA5 pathway, as well as PKR. Alternatively, derepression of these elements may induce transcription replication conflicts (TRCs), resulting in R-loop formation. R-loops can lead to DNA damage, and/or activate the cGAS/STING pathway. Both the MAVS pathway and the cGAS/STING pathway converge to activate type I interferon (IFN) responses, resulting in decreased cell fitness and/or increased immunogenicity.

We do agree with the assessment that the study would be strengthened by in vivo studies. However, there are 4 different isoforms of KDM5 (3 in females), and existing KDM5specific inhibitors do not have adequate PK/PD properties for in vivo studies. We would also like to note that most mouse studies have not been proven to accurately predict immunotherapy responses in patients. Future studies in ex vivo tumor models would strengthen the clinical relevance of these studies. In the interim, we have added some normal macrophage studies in Figure S5 and an example of studies in normal T-cells below. Such studies will also be important to ensure that future KDM5 inhibitors do not have adverse effects on the immune system. Here, we observe that KDM5 inhibition appears to have neutral or slightly reduced T cell viability with KDM5 inhibition (Author response image 2a). However, KDM5 inhibition also results in increased CD107a expression in T-cells, indicative of a more cytotoxic phenotype (Author response image 2b). These studies suggest that KDM5 inhibitors do not have significant adverse effects on T cells or macrophages (figure S5) in the normal immune environment.

**Author response image 2. sa3fig2:** KDM5 inhibition does not have significant adverse effects on T-cells. a) Fold change proliferation of T-cells from 2 different human donors (left and right panels on graph) activated with 0.25ug/ml CD3 and treated with the indicated concentrations of C48 or a positive control (CBLB) compared to vehicle controls. b. FACS plots and histograms of CD107a surface expression (x-axis) versus forward scatter (FSC, y-axis) of T-cells from 2 different humans donors activated with 0.25ug/ml or 0.5mug/ml CD3 and treated with the indicated concentrations of C48.

**Specific comments and answers to Reviewer #1:**

We have added some additional analysis of data from other breast cancer cell lines to strengthen our points (Figure S2f, Figure S3e, Figure S4g-h, k.) We have also uploaded all the data to Geo with the following accession numbers :

GSE296387: H3K4me3 CUT-and-Tag data

GSE296584: S9.6 CUT-and-Tag data

GSE296974: RNA-sequencing data

**Responses to Reviewer #1 (Recommendations for the authors):**

(1) We have not conducted genomic studies comparing KDM5 expression to retroelement activation status in the tumor data sets but recognize that this is important for future studies. Again, there are several KDM5 isoforms and looking at repeat expression in these larger data sets is complex. We have added some data correlating KDM5 expression with ISG signatures in Figure S3j-l as well as in the graph below (Author response image 3). The correlation with ISG and AP signatures is modest, but strongest for KDM5B and C in breast cancer data sets, consistent with our disruption data for these 2 isoforms. As mentioned above, we do agree that future studies of KDM5s along with a broader analysis of other epigenetic modifying enzymes over repeats in various cancer types will shed light on the role of histone modifying enzymes in suppressing “viral mimicry” in tumors.

**Author response image 3. sa3fig3:** Correlation between gene expression and IFN gene set GSVA scores in breast cancer cell lines. a) Pearson correlation score between gene expression and IFN signature (ISG) gene set variation analysis (GSVA) scores in breast cancer cell lines as reported in DepMap. Higher ranks indicate an inverse correlation between expression of the individual gene and the expression of the ISG gene set. Correlation ranks for KDM5A, B and C are highlighted. b) as in a), but comparing gene expression to antigen presentation (AP) GSVA scores.

(2) We apologize for the mislabeling in figure 2B – has been corrected in the revised version.

(3) We agree that blocking the cGAS/STING pathway, only partially rescues the ISREGFP and HLA-A, B, C phenotype in HCC1428 cells. We have added data (Figure S2f) showing that this rescue is stronger in MCF7 cells. It is possible that the MDA5/MAVS pathway may also contribute to activation of the Type I interferon response. However, we have data that MAVS plays a minor (if any) role in this context, as MAVS KO minimally decreases C48-induced ISRE-GFP activity and HLA-A, B, C surface expression in HCC1428 cells (added Figure S2g).

Furthermore, there is no significant increase in dsRNA observed (using J2 antibody as a readout in immunofluorescence experiments) with C48 treatment as compared to 5’-azacytidine treatment or ADAR K/O (data not included). However, we have not performed MAVS/PKR K/O experiments to completely rule out the involvement of the dsRNA sensing pathways.

(4) These experiments were performed in the operetta imaging system, rather than confocal imaging, and therefore we do not have such images. Quantification of RNaseH1-GFP in the whole cell is reported in the figure, as RNaseH1-GFP signal is increased in both the nucleus and the cytoplasm with C48 treatment. This is not unexpected, as our data suggest that R-loop formation occurs in repetitive regions of the genome that are de-repressed by KDM5 inhibition in the nucleus, and the RNA/DNA hybrids, generated from R-loops, may activate cGAS/STING pathway in the cytoplasm.

(5) Disruption of siXPF and siXPG is relatively toxic in itself. Complete knockouts in breast cancer cells were not viable and we partially knocked down XPF using siRNA instead. We do agree that these kinds of rescue studies need to be expanded upon in future studies, but they served as further proof of the conclusions presented here.

(6) We have provided all the data in Geo and alternative representations can be made.

(7) Unfortunately, CUT-and-Tag experiments were not performed in cells expressing siXPF and therefore we cannot provide this data. However, XPF has been previously shown to be responsible for excising R-loops from the genome, rendering them detectable by cGAS/STING in the cytoplasm (Crossley et al, 2022, referenced in the current MS). Therefore, while we demonstrate that XPF knockdown attenuates type I IFN pathway activation upon KDM5 inhibition, it may not necessarily reduce R-loop formation in retroelements; it may just prevent their excision and downstream cGAS/STING activation. We do agree that CUT-and-Tag experiments in cells treated with siXPF versus siControl will have to be performed in the future to test this hypothesis.

**Responses to Reviewer #2 (Recommendations for the authors):**

(1) We have modified the text as well as the figure legend to state that this is a simplistic representation of the pathway in normal cells. As stated in the introduction, these pathways can be modified in tumors. The data presented suggest that the dsRNA pathway can be activated in all breast cancer cell lines tested, whereas more variation is observed in the activation of the STING pathway.

(2) The ADAR guides target ADAR 110 and p150 but not ADAR2. This has been clarified in the text.

(3) The guides have been renamed in the figure as the reviewer suggests.

(4) It has been shown by others that KDM5 can occupy the STING promoter (https://pubmed.ncbi.nlm.nih.gov/30080846/); which supports the reviewer’s suggestion that STING upregulation in HMECs may be due to increased H3K4me3 at the STING gene. However, we argue that STING upregulation is not sufficient to activate “viral mimicry” due to the absence of “tumor-specific R-loops” (due to an increase in TRC in tumor cells) in normal cells. It is interesting to note that the S9.6 signal in subtelomeric regions is increased in HMECS similar to what is observed in tumor cells. However, the S9.6 signal over other repeats is not (Author response image 4), suggesting that C48-induced increases over non-telomeric repeats are tumor specific. This suggests that the tumor-specific increases in R-loop formation, which lead to “viral mimicry” activation, are not driven by those formed in subtelomeric regions. Future studies will have to expand on these findings.

**Author response image 4. sa3fig4:** Percent of S9.6 reads that align to repetitive genome in HMEC cells. (a) % of total aligned S9.6 reads that map to subtelomeric region in HMEC cells treated with DMSO or 2.5 μM C48. (b) % of total aligned S9.6 reads that map to repetitive elements in general in HMEC cells treated as in a.

(5) Clarity on R-loop quantification has been added to the figure legend as well as in the Materials and Methods section. Mean fluorescence intensity in the whole cell (this includes both nuclear and cytoplasmic signals) was quantified together and normalized to the number of DAPI-stained nuclei per well. As mentioned above all quantified in the Operetta imaging system.

(6) We have added some data that shows that increases in H3K4me3 is observed in and around ISGs upon KDM5 inhibition (Figure S4f). However, without time course experiments it is difficult to assess whether these are direct effects of the KDM5 inhibitor or indirect effects from activation of Type I IFN (similarly to what has previously been reported with 5’-azacytidine induction of “viral mimicry”, https://pubmed.ncbi.nlm.nih.gov/26317465/).

(7) We have previously included data showing that S9.6 reads in repeats that do not display C48-mediated increases in H3K4me3 also do not increase with C48 treatment (this is now Figure S4o). In addition, we have added some data showing that repeats with increased H3K4me3 and repeats with increased transcription upon C48 treatment also have increased S9.6 reads. Repeats that display both increases in H3K4me3 and mRNA expression have even greater increases in S9.6 signal compared to repeats that have increases in either one (Figure S4m-n). Taken together, this data suggest that KDM5 inhibition increases H3K4me3 in repeats, thereby allowing for their transcription, which can increase the probability of Transcription replication conflicts (TRC) and R-loop formation at such loci.

(8) As mentioned earlier in this response, while we observe increased S9.6 reads in subtelomeric regions of HCC1428 cells upon KDM5 inhibition, we also observe this in normal HMEC cells. Since KDM5 inhibition does not induce viral mimicry in HMEC cells, this suggests that R-loops formed in subtelomeric regions do not dictate the response observed with C48 treatment in breast cancer cells.

We hope that these answers to the reviewers comments as well as the additional data provided strengthens our findings.